# Promotion of Hair Regrowth by Transdermal Dissolvable Microneedles Loaded with Rapamycin and Epigallocatechin Gallate Nanoparticles

**DOI:** 10.3390/pharmaceutics14071404

**Published:** 2022-07-04

**Authors:** Yali Lin, Ruomei Shao, Tong Xiao, Shuqing Sun

**Affiliations:** 1Institute of Biopharmaceutical and Health Engineering, Shenzhen International Graduate School, Tsinghua University, Shenzhen 518055, China; lin-yl19@mails.tsinghua.edu.cn (Y.L.); shruomei@aliyun.com (R.S.); xiaot21@mails.tsinghua.edu.cn (T.X.); 2Department of Biomedical Engineering, Tsinghua University, Beijing 100084, China; 3Department of Chemical Engineering, Tsinghua University, Beijing 100084, China

**Keywords:** hair regrowth, microneedles, nanoparticles, rapamycin

## Abstract

Interest in transdermal delivery methods for stimulating hair regrowth has been increasing recently. The microneedle approach can break the barrier of the stratum corneum through puncture ability and improve drug delivery efficiency. Herein, we report a dissolvable microneedle device for the co-delivery of rapamycin and epigallocatechin gallate nanoparticles that can significantly promote hair regeneration. Compared with the mice without any treatment, our strategy can facilitate hair growth within 7 days. Higher hair shaft growth rate and hair follicle density with inconspicuous inflammation were exhibited in C57BL/6 mice, elucidating its potential for clinical application.

## 1. Introduction

Hair loss, a common and distressing symptom, has become a condition that plagues billions of people worldwide [1,2]. Currently, topical minoxidil [3], oral finasteride [4], and hair follicle transplantation [5] are common clinical treatments but still face challenges such as uncontrolled adverse effects, high cost, poor compliance of patients, and time commitment [6,7]. Although other pharmacological treatments, such as valproic acid [8], Ru58841 [9], etc., as well as phytochemicals from botanicals [10], were demonstrated to promote hair growth, most drug formulations are prepared for topical administration [11]. However, because of the protective barrier of the stratum corneum, the percutaneous absorption of drugs is hindered, resulting in low percutaneous permeability and poor treatment compliance [12,13]. Therefore, effective treatments are urgently needed.

Transdermal delivery systems are required for successful therapeutic application beyond conventional venous injections and oral or topical administration [14,15,16,17]. Microneedles (MNs) with three-dimensional microstructures with microscale lengths (usually less than 1000 μm) are a potential direction, which can overcome the complex and dynamic barriers of the stratum corneum, control the delivery dose, and improve the efficiency and accuracy of drugs [16,18]. In recent years, microneedle drug delivery systems therapy for alopecia disease has received extensive attention and fueled the hair loss therapy research field [19,20,21,22]. However, individual differences have led to an ever-increasing need for precise hair growth.

In this study, a dissolvable microneedle device to deliver nanoparticles (NPs) was designed (Figure 1). Polyvinylpyrrolidone (PVP), with good biocompatibility, mechanical properties, and water-soluble properties, served as a microneedle structure [23]. When the microneedle penetrated into the skin, it dissolved rapidly and released nanoparticles, enabling controlled and sustained release of drug nanoparticles into the skin. Rapamycin (RAPA) and epigallocatechin gallate (EGCG) were selected as target drugs. Rapamycin, an autophagy-activating small molecule, can induce the transition of the hair cycle from the telogen to anagen phase, thereby promoting hair regrowth [24,25]. The nanoparticles were formed by polylactic–glycolic acid copolymer (PLGA) with encapsulation of rapamycin (RAPA-PLGA NPs), which helps to improve the sustained-release effect and prolong the incubation period of rapamycin [26]. In addition, EGCG has antioxidation and anagen activation effects [27]. Keratin (KE) is the main component of hair [28]. Crosslinking keratin with EGCG to form nanoparticles (EGCG-KE NPs) can not only provide nutrition for hair but also benefit the proliferation of hair follicle cells. From a series of in vitro and in vivo experiments, we demonstrated that the RAPA-PLGA and EGCG-KE NP-loaded MN device (DMN) can promote hair regeneration within as few as 7 days.

## 2. Materials and Methods

### 2.1. Materials and Animals

Poly(lactic-co-glycolic acid) (PLGA, 50/50), rapamycin (RAPA, 98%), Keratin(KE) solution (5 mg/mL, Aladdin, Shanghai, China), epigallocatechin gallate (EGCG, 95%, Aladdin, Shanghai), polyvinyl pyrrolidone (PVP, Mw 400,000), acetone, dithiothreitol (DTT), polyethylene-polypropylene glycol (F68), polydimethylsiloxane (PDMS, Dow Corning Sylgard 184), formaldehyde, C57BL/6 mice (6 weeks old, Guangdong Medical Laboratory Animal Center). All mouse experiments were approved by the Animal Care Committee of Tsinghua University.

### 2.2. Preparation of Nanoparticles

RAPA-PLGA NPs: 10 mg of PLGA and 1 mg of rapamycin were dissolved in 1 mL of acetone solution. PLGA solution was added dropwise to 10 mL of 0.1% F68 solution with stirring at room temperature overnight. After centrifugation at 10,000 r/min for 20 min, the collected nanoparticles were dispersed in pure water and stored at 4 °C.

EGCG-KE NPs: 10 mL of keratin solution (1 mg/mL) was reduced by 20 μg of DTT for 4 h, and then 40 mL of EGCG solution (5 mg/mL) was added. Then, 100 μL of 37% formaldehyde solution was added dropwise into the mixture with stirring at room temperature for 12 h. After centrifugation at 10,000 r/min for 20 min, the collected nanoparticles were dispersed in pure water and stored at 4 °C.

### 2.3. Fabrication of DMN

A 10 × 10 array microneedle (a round base diameter of 320 μm, a height of 680 μm in each microneedle, 600 μm of tip–tip spacing) anode model was prepared by Suzhou Machinery (China). Then, the PDMS cathode mold was prepared by the template method. PVP solution (1 g/mL) mixed with the drug (RAPA-PLGA NPs and/or EGCG-KE NPs) was added to the PDMS microneedle mold and then vacuumed for 2 min. The excess PVP solution was removed using a spin coater. Then, 13% PVA/sucrose solution was added as the MN substrate. The whole model was placed in a desiccator for 2 days until completely dry. The microneedle patch was peeled off from the PDMS model and stored in a desiccator.

An example of how to prepare a MN patch with 1 μg of RAPA is shown. Next, 1 g of PVP was mixed with 1 mL of RAPA-PLGA NPs solution (containing 1 mg RAPA encapsulated in NPs). Then, the microneedle wells (2 μL/MN patch) were filled with PVP solution (0.5 mg/mL RAPA) by vacuum. The excess PVP solution was removed by a spin coater to ensure wells contained only drugs. Thus, the microneedle wells contained 1 μg (2 μL × 0.5 mg/mL) of RAPA. Different dosage amounts of model drugs in MN were controlled in the same way.

### 2.4. Characterizations of NPs and DMN

Ultraviolet spectroscopy (UV): the aqueous solution of the samples (nanoparticles, KE, rapamycin, EGCG) was measured by an ultraviolet spectrophotometer in the wavelength range of 200–400 nm.

Dynamic light scattering (DLS): the nanoparticles were ultrasonically dispersed in pure water, and their size and zeta potential were obtained using a Malvern Mastersizer 3000 analyzer.

Scanning electron microscopy (SEM): the lyophilized nanoparticle powder or MN was placed in a silicon wafer. The surface of the sample was sprayed with gold, and the morphology of the nanoparticles was observed under a scanning electron microscope (5 kV).

Mechanical strength test: the microneedle was pressed by the top sensor at a speed of 0.1 mm/min until destroyed, and the force-displacement data were recorded by an electromechanical universal testing machine.

In vitro drug release study: a vertical Franz diffusion was used for the PLGA nanoparticle release study. The upper diffusion pool was added to the nanoparticle dispersion, and the lower pool was filled with 0.1% F68 PBS solution (pH = 7.4). A filter membrane (0.2 μm) was located between the diffusion pool and the supply pool. The Franz diffusion cell was stirred at 37 °C. One milliliter of solution from the supply pool at the setting time was used to detect the concentration of rapamycin by UV.

### 2.5. Animals Experiments

Rapamycin MN: following anesthesia by 1.25% tribromoethanol, dorsal skin hair (an area of approximately 2 cm × 2 cm) was shaved using an electric hair clipper and hair removal cream. Six-week-old C57BL/6 mice with MN administration were treated on the first and seventh day. The microneedle was pressed into the dorsal skin with the thumb for 30 s, followed by fixation with medical tape paste and the microneedle base was peeled off after 2 h. In rapamycin MN groups, dosage from low to high (0.001 μg, 0.01 μg, 0.1 μg, 1 μg, 10 μg rapamycin/MN patch) was applied to the treatment area. In the topical dosing groups, 30 μL of rapamycin lecithin gel (low dose to high dose: 0.2 μM, 2 μM, 20 μM) was applied to the treatment area (2 cm × 2 cm) every other day. The shaved untreated mice served as controls.

DMN: following anesthesia by 1.25% tribromoethanol, the dorsal skin hair (an area of approximately 2 cm × 2 cm) was shaved by using an electric hair clipper and hair removal cream. Six-week-old C57BL/6 mice with MN administration were treated on the first and seventh day. The microneedle was pressed into the dorsal skin with the thumb for 30 s, followed by fixation with medical tape paste, and the microneedle base was then peeled off after 2 h. In the topical dosing group, 30 μL of rapamycin lecithin gel (2 μM) or 20 μL of 2% EGCG ethanol solution was applied on the treatment area (2 cm × 2 cm) every other day. The shaved untreated mice served as controls.

### 2.6. Histology and Immunofluorescent

At the time point, mice were sacrificed, and skin samples were collected from the back. Samples were fixed with 4% paraformaldehyde solution for 24 h and then dehydrated and embedded in wax using an automatic tissue dehydrator. After sectioning, the samples were subjected to hematoxylin and eosin (H&E) staining. H&E staining images were collected by optical microscopy.

For immunofluorescence, the sample slices were immersed in 0.01 M Tris-EDTA solution for antigen retrieval and then blocked in 5% goat serum. The samples were incubated with primary antibody-targeting (CD3, CD68; 1:100; Bioscience) working solution for 12 h at 4 °C, and then washed with PBS three times. For secondary antibody incubation, the sections were stained with FITC- and rhodamine-conjugated secondary antibodies, and counterstained with DAPI. The fluorescent images were collected by fluorescence microscopy.

### 2.7. Western Blotting

Skin tissue samples from the shaved dorsal area were lysed with lysis buffer for protein extraction. Next, equal amounts of protein were separated on SDS–PAGE gels by electrophoresis (100 V, 120 min) and transferred to nitrocellulose membranes (300 mA, 40 min). The protein membrane was rinsed and blocked with BSA solution at room temperature for 60 min. Next, the membrane was incubated with primary antibodies (β-catenin, LC3, P62, AKT, p-AKT) at 4 °C overnight and then incubated with HRP-labeled secondary antibody for 1 h at room temperature.

### 2.8. Statistical Analysis

All experiments were independently repeated at least 3 times (*n* ≥ 3). The results are presented as the mean ± standard deviation. Data analysis was processed by SPSS and GraphPad software. A probability value (*p* < 0.05) by *t*-test was considered significant (* *p* < 0.05, ** *p* < 0.01, *** *p* < 0.001).

## 3. Results and Discussion

### 3.1. Characterization of MNs and Nanoparticles

The microneedles were prepared using a two-step template method (Figure 2A). First, PVP-based microneedles were prepared, and then PVA/sucrose was covered as a flexible substrate. The finished microneedle patch was a 10 × 10 microneedle array on a 1 cm × 1 cm base. Scanning electron microscopy (Figure 2B,C) and laser confocal microscopy (Figure 2D) showed that the DMN was smooth and uniform, maintaining a good conical shape and sharp needle tip. The base diameter of each microneedle was 320 μm, and the height was 680 μm. Since the epidermal layer of skin is the main barrier for transdermal drug delivery, studies discovered that a force of at least 0.1 N/needle is required to penetrate the skin [29]. According to the mechanical strength test, the force–displacement curve of the microneedle showed linear continuity, indicating that the PVP microneedles have good mechanical properties and are not easy to destroy during the compression process (Figure 2E). When half of the needle length (300 μm) was compressed, the mechanical strength of a single microneedle was 300 mN. This demonstrated that the microneedle can satisfy the mechanical conditions for penetrating the skin.

In this study, EGCG and RAPA were used as drugs to activate hair follicle regeneration. Rapamycin nanoparticles were prepared using the emulsification and volatilization method (oil-in-water O/W method) to form microemulsions dispersed in water, which is a common synthetic method for biodegradable polymer nanoparticles. Hydrophobic rapamycin was encapsulated in the PLGA polymer. In contrast, EGCG nanoparticles were formed by chemical cross-linking with keratin, and the formation mechanism was the polycondensation reaction of methyl of KE and the thiol groups (-SH) of EGCG [30]. To characterize the physicochemical properties of the nanoparticles, scanning electron microscopy showed that both NPs were spherical, with good dispersibility and good singleness (Figure 3A,B). DSL illustrated that the average diameters of PLGA nanoparticles (Figure 3E) and EGCG nanoparticles (Figure 3F) were 150 nm and 100 nm, respectively. Notably, the UV absorption peaks of rapamycin and EGCG were 280 nm and 275 nm, respectively. The synthesized nanoparticles (Figure 3C,D) also had the same absorption peak, indicating that the optical properties of the drugs were not affected during the nanoparticle formation process. In addition, the loading capacity is an important index to evaluate the quality of nanoparticles. The loading capacities of RAPA and EGCG NPs were 82% and 76%, respectively. The in vitro drug release curve (37 ℃, pH = 7.4) demonstrated the sustained-release effect of PLGA NPs, with a slow release of 70% RAPA in 13 days (Figure 3G).

### 3.2. Transcutaneous Permeation Analyses

To evaluate the transcutaneous permeation and biocompatibility of microneedles, a series of in vitro and in vivo experiments was performed. When the microneedle pierced the mouse dorsal skin for 30 s followed by 5 min of waiting, H&E staining demonstrated that the microneedle created a microchannel, confirming the transdermal function of the microneedles (Figure 4A). The deepest penetration depth could reach around 480 μm. The average insertion depth of MN was around 400~450 μm because of skin natural elasticity. Since the skin thickness on mice was 300~700 μm through the hair cycle, the depth of pore formation made it easy and acceptable for drugs to deliver into hair follicles [31]. At the same time, the MN base could be peeled off the skin after 2 h, leaving the microneedles dissolved in the skin. To verify the solubility of the microneedles, Figure 3B illustrates that the dissolution degree of PVP microneedles pierced into porcine cadaver skin was higher with a longer residence time. The microneedles were completely dissolved in 60 s, proving the high solubility of PVP microneedles. In clinical safety experiments, after microneedles were pierced into mice, microneedle patterns appeared in the dorsal skin and the treated skin recovered within 60 min (Figure 4C). To further verify the cause of inflammation, we evaluated the biosafety of the DMN group, the PVP microneedle group, and the control group by immunofluorescence staining. CD68 is a marker for macrophages, and CD3 is a marker for T cells. Figure 3D illustrates that CD68 and CD3 were not significantly expressed in all groups and indicated no inflammatory reaction, proving the biocompatibility of DMN.

### 3.3. Hair Regrowth Evaluation

C57BL/6J mice with obvious hair cycling are known to enter a prolonged telogen phase by six weeks of age [32]. Thus, hair-shaved mice were commonly used as animal models for hair regrowth evaluation to induce the transformation of the hair cycle. In our study, rapamycin dosage can affect the process of hair regeneration. Therefore, we first determined the dosage of rapamycin nanoparticles loaded in microneedles. Forty-two-day C57BL/6 mice were divided into nine groups (mice number: *n* = 5/group): five microneedle administration groups from low dose to high dose, three topical administration groups with different dosages, and a control group without any treatment. Figure 5A shows representative images of mice during the treatments, and Figure 5B illustrates the hair loss therapy in each group. Hair regrowth was observed on day 10 after shaving when the dosage of rapamycin was 0.01 μg~1 μg/DMN. The longest hair shaft length was obtained in mice treated with 0.1 μg of rapamycin (Figure 5C). In contrast, the control group without treatment showed almost no hair growth. Western blotting (Figure 5D) confirmed that rapamycin could induce autophagy. β-Catenin is a marker of hair follicle cycle activation, while p62 and LC3 are autophagy marker proteins that regulate hair follicle morphogenesis. Figure 5D demonstrates that the protein expression levels of β-catenin, p62 and LC3 with low-dose rapamycin were much higher, which further illustrated that low-dose rapamycin can effectively promote hair regeneration.

Compared with topical administration, a reduced dosage of drugs is used via the MN delivery strategy. Our DMN patch was loaded with 0.1 μg of rapamycin and 4 μg of EGCG. To evaluate the therapeutic effect, mice were divided into six groups, and photographs of hair growth states at each time point are shown in Figure 6A. A faster hair growth rate was observed in the rapamycin administration groups (H1, H2, H4), while a higher hair follicle density was shown in the EGCG administration groups (H1, H3, H5) (Figure 6B,C). H&E staining of longitudinal and transverse sections of mouse skin was conducted to reveal hair regeneration behavior (Figure 6E). In general, compared with other groups, DMN for the codelivery of RAPA and EGCG generated the best therapeutic effect on hair regeneration. Hair in the DMN region was significantly the longest and densest. Western blotting further indicated higher expression levels of β-catenin (hair follicle regulatory protein) and p-AKT (a marker of hair follicle cells for proliferation and anti-aging) via DMN dorsal skin to activate hair follicles to enter a new hair cycle (Figure 6D).

## 4. Conclusions

In summary, a dissolvable PVP-based microneedle patch was prepared for the codelivery of RAPA and EGCG nanoparticles. Microneedles were constructed with high mechanical properties to break the barrier of the stratum corneum via punctuation and then dissolve rapidly, transporting RAPA and EGCG nanoparticles to the hair follicle niche. In vivo experiments demonstrated that the DMN can significantly improve hair regrowth with biocompatibility. Our findings indicate that it is expected to be utilized as a potential candidate to address hair loss in a minimally invasive manner.

## Figures and Tables

**Figure 1 pharmaceutics-14-01404-f001:**
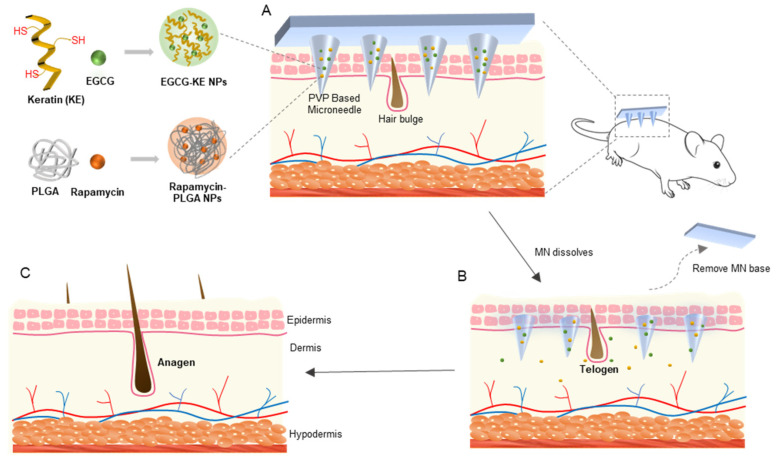
Schematic of the microneedle device for hair regrowth. (**A**) Our DMN (PVP-based microneedle loaded with RAPA-PLGA and EGCG-KE NPs) penetrate the skin. (**B**) The microneedles dissolve rapidly and release nanoparticles and drug transport into the hair follicle niche. (**C**) Drugs can activate the healthy hair cycle, accelerate the transition of hair follicles from telogen to anagen, and promote hair regrowth.

**Figure 2 pharmaceutics-14-01404-f002:**
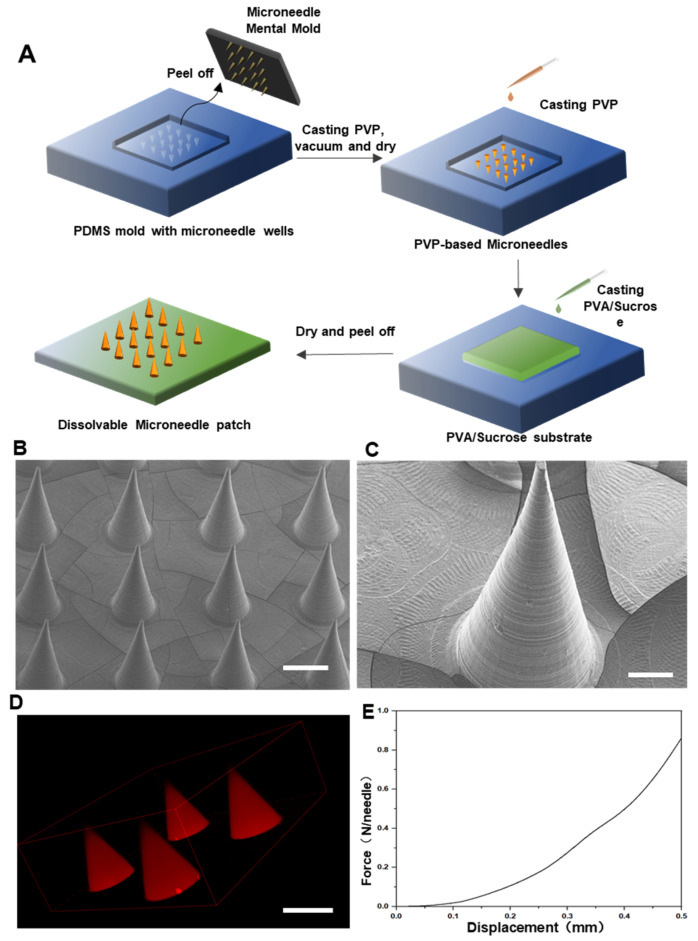
Characterization of MNs. (**A**) Preparation scheme of a drug NP-loaded MN. (**B**) SEM images of MNs. Scale bar: 300 μm. (**C**) SEM images of MNs. Scale bar: 100 μm. (**D**) Laser confocal microscopy image of MNs. Scale bar: 300 μm. (**E**) Mechanical strength of MNs and photograph after compression.

**Figure 3 pharmaceutics-14-01404-f003:**
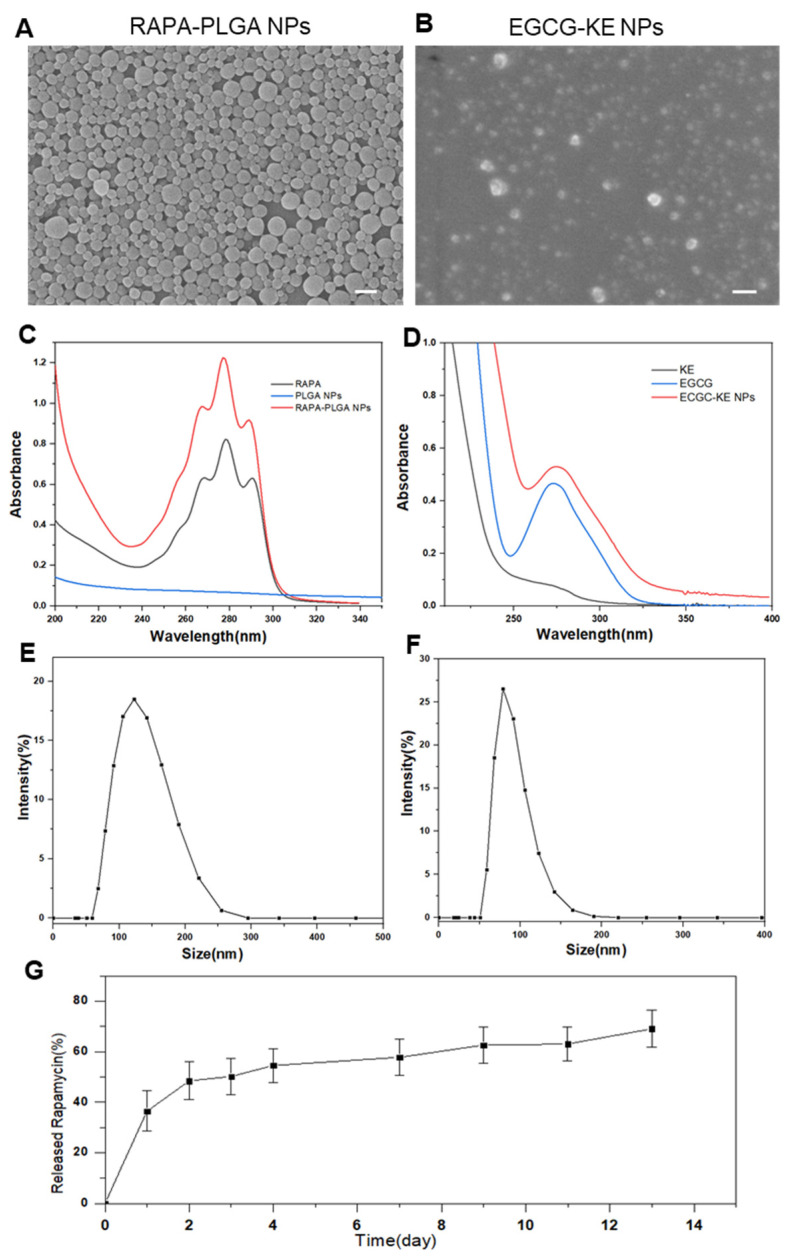
Characterization of NPs. SEM image of PLGA-RAPA NPs (**A**) and EGCG-KE NPs (**B**). Scale bar: 200 nm. UV–vis of PLGA-RAPA NPs (**C**) and EGCG-KE NPs (**D**). Particle size distribution with a photograph of PLGA-RAPA NPs (**E**) and EGCG-KE NPs (**F**) in water. (**G**) In vitro release test of RAPA-PLGA NPs (*n* = 3, mean ± SD).

**Figure 4 pharmaceutics-14-01404-f004:**
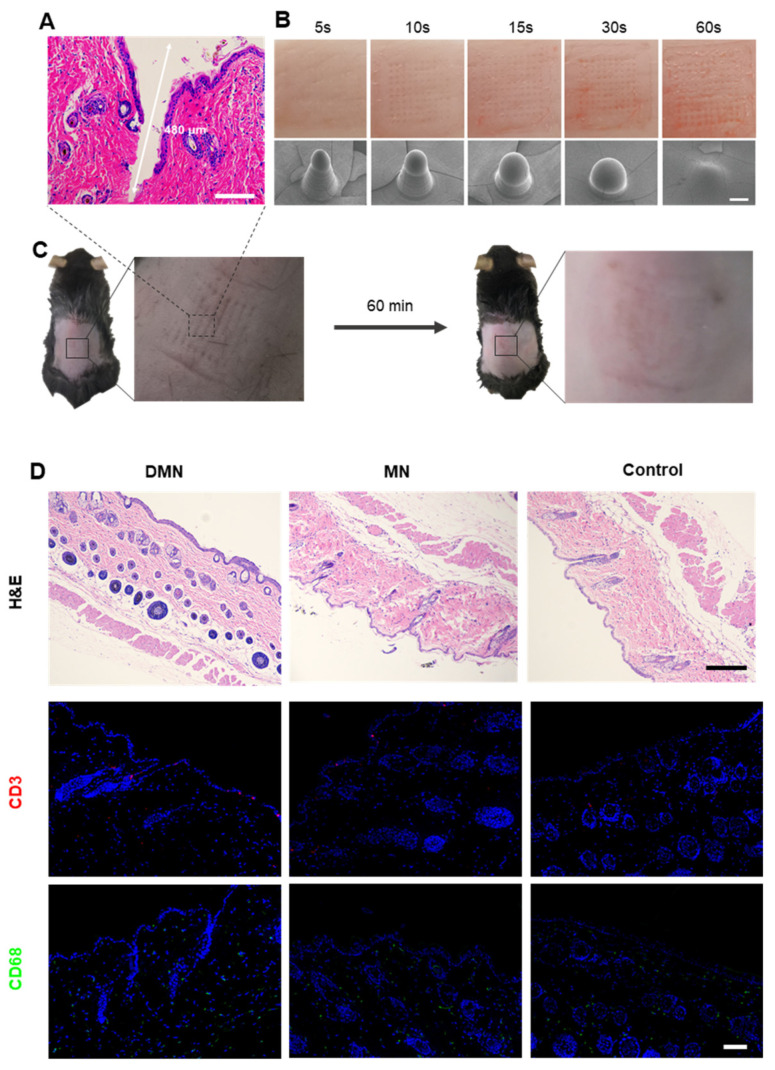
In vitro and in vivo administration of MNs. (**A**) H&E staining image of the mouse dorsal skin after MN penetration. Scale bar: 100 μm. (**B**) Photographs of porcine cadaver skin and SEM images of MNs after penetration at different time points. Scale bar: 100 μm. (**C**) Photographs of mice after penetration and at 60 min postinsertion with removal of MNs. (**D**) H&E and immunofluorescence staining of treated mouse skin at day 10. CD3 (marker of lymphocyte infiltration) is shown in red, CD68 (marker of macrophages) is shown in green, and cell nuclei (DAPI) are shown in blue. Scale bar: 50 μm.

**Figure 5 pharmaceutics-14-01404-f005:**
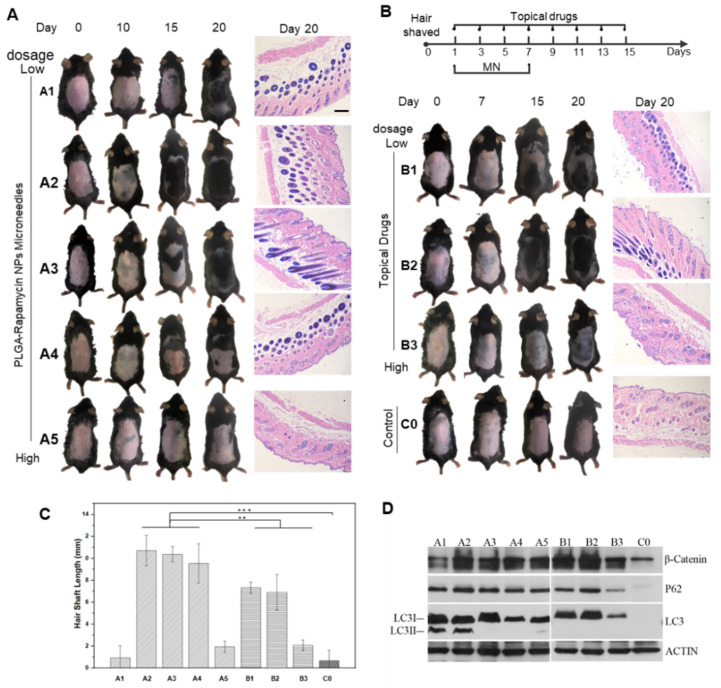
In vivo hair regrowth study by treatments with different dosages of rapamycin. (**A**) Schematic illustration of the hair loss treatments. (**B**) Physical and H&E staining images (scale bar: 100 μm) of mice treated with RAPA-PLGA NP-loaded MNs and topical RAPA. The untreated mice served as a control. (**C**) Hair shaft analysis chart of new hair grown on treated mouse dorsal skin (*n* = 3, mean ± SD). (**D**) Western blotting assay of the expression level of proteins related to hair follicle growth activation (β-Catenin) and autophagy (p62 and LC3). A probability value (*p* < 0.05) by *t*-test was considered significant (** *p* < 0.01, *** *p* < 0.001).

**Figure 6 pharmaceutics-14-01404-f006:**
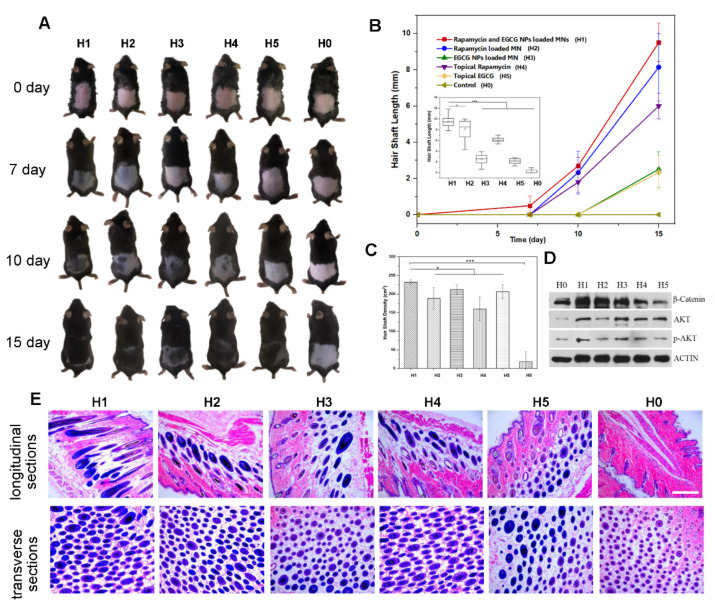
In vivo study of DMN for hair regrowth. (**A**) Physical images of mice treated with RAPA-PLGA and EGCG-KE NP-loaded MNs (H1), RAPA-PLGA-loaded MNs (H2), EGCG-KE NP-loaded MNs (H3), topical RAPA (H4), and topical EGCG (H5). Untreated mice (H0) served as a control. (**B**) Hair shaft analysis chart of new hair grown on treated mouse dorsal skin as a function of treatment time and day 15 (*n* = 3, mean ± SD). (**C**) Hair follicle density analysis chart of new hair grown on treated dorsal skin at day 15 (*n* = 3, mean ± SD). (**D**) Western blotting assay of the expression level of proteins related to hair follicle growth activation (β-Catenin, p-AKT). (**E**) H&E staining images of longitudinal sections (top) and transverse sections (bottom) of mouse dorsal skin (scale bar: 100 μm). A probability value (*p* < 0.05) by *t*-test was considered significant (* *p* < 0.05, *** *p* < 0.001).

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
