# Peer review of "Promotion of Hair Regrowth by Transdermal Dissolvable Microneedles Loaded with Rapamycin and Epigallocatechin Gallate Nanoparticles"

_pharmaceutics, 2022, doi:10.3390/pharmaceutics14071404_

Round 1

Reviewer 1 Report

Lin and co-workers did a good job designing and conducting the experiments involving dissolvable microneedles (MNs)for the promotion of hair growth. However, please address the following comments:

1) Please further justify the use of mice model for conducting MNs based Transdermal systems.

2) The depth of the MNs was around 500nm, please comment on the impact of inflammation caused by such a deep insertion, especially considering very thin skin thickness on mice.

3) Please provide any references for the same, justifying the depth of pore formation is acceptable.

4) Please provide any quantitative determination of model drug in the skin layers or systemic uptake after MNs insertion. 

Author Response

Please see the attachment (Reviewer 1).

Reviewer 2 Report

Congratulations for the hard and excellent work.

I have minor suggestions:

the author should revised references and spacing in some sentences.

example:

line 20 - (...)"oral finasteride4" should be written as "oral finasteride[4]" ; line 21 - (...)"hair follicle transplantation5" should be written as "hair follicle transplantation[5]" , etc.

In 2.1.Materials and methods section, the author only writes the acronym KE without full meaning (most likely keratin (KE))

In 2.2. Preparation of nanoparticles section, the author writes: "RAPA-PLGA NPs: 10 mg of PLGA and 1 mg of rapamycin were dissolved in 1 ml of acetone solution. PLGA solution was added dropwise to 10 ml of 0.1% F68 solution with stirring, and then the solution was rotated at room temperature overnight.

I suggest rewriting as: "RAPA-PLGA NPs: 10 mg of PLGA and 1 mg of rapamycin were dissolved in 1 ml of acetone solution. PLGA solution was added dropwise to 10 ml of 0.1% F68 solution with stirring, at room temperature overnight.

Author Response

Comment: 1. The author should revised references and spacing in some sentences.example: line 20 - (...)"oral finasteride4" should be written as "oral finasteride[4]" ; line 21 - (...)"hair follicle transplantation5" should be written as "hair follicle transplantation[5]" , etc.

Response: Thanks for raising this point. We have checked the entire manuscript carefully and revised references to follow the guidelines of the journal.

Comment: 2. Materials and methods section, the author only writes the acronym KE without full meaning (most likely keratin (KE))

Response: We thank the reviewer for the close read and suggestions. We have added the full meaning of KE as followed: “Materials and animals: Poly(lactic-co-glycolic acid) (PLGA, 50/50), rapamycin (RAPA, 98%), Keratin(KE) solution (5 mg/ml, Aladdin, Shanghai), epigallocatechin gallate (EGCG, 95%, Aladdin, Shanghai), polyvinyl pyrrolidone (PVP, Mw 40000)”

Comment: 3. Preparation of nanoparticles section, the author writes: "RAPA-PLGA NPs: 10 mg of PLGA and 1 mg of rapamycin were dissolved in 1 ml of acetone solution. PLGA solution was added dropwise to 10 ml of 0.1% F68 solution with stirring, and then the solution was rotated at room temperature overnight.

I suggest rewriting as: "RAPA-PLGA NPs: 10 mg of PLGA and 1 mg of rapamycin were dissolved in 1 ml of acetone solution. PLGA solution was added dropwise to 10 ml of 0.1% F68 solution with stirring, at room temperature overnight. "

Response: We are glad to accept your suggestion. The preparation of nanoparticles section has been revised.

Reviewer 3 Report

Comments and Suggestions for Authors

In the present manuscript, the authors reported the interest in transdermal delivery methods for stimulating hair regrowth has been increasing recently. The microneedle approach can break the barrier of the stratum corneum through puncture ability and improve drug delivery efficiency. Herein, we report a dissolvable microneedle device for the codelivery of rapamycin and epigallocatechin gallate nanoparticles that can significantly promote hair regeneration. Compared with the mice without any treatment, our strategy could facilitate hair growth within 7 days. A higher hair shaft growth rate and hair follicle density with inconspicuous inflammation were exhibited in C57BL/6 mice, elucidating its potential for clinical application. The presented research includes both fundamental experimental and application studies, which are essential for substantiating the claim with solid data and, as a result, paving the way for the use of this nanostructure in the field of innovative materials for skin treatment..

The authors should pay more attention on the following points.

1. Correct the in text references throughout the paper.

2. Higher doses of EGCG are hepatotoxic, if 750 mg/kg dosed twice a day (doi: 10.1016/j.fct.2009.10.030). Daily intake of 800 mg or more could increase risk of liver damage (doi:10.2903/j.efsa.2018.5239).  Did you consider the toxicity concentration levels of EGCG and the dosing amount when designing this study?

3. In section 2.3, it is mentioned that the PDMS cathode mold is prepared by the template method. PVP solution (1 g/ml) mixed with the drug (RAPA-PLGA NPs and/or EGCG-KE NPs) was added to the PDMS microneedle mold and then vacuumed for 2 min. The excess PVP solution (assuming it contains RAPA-PLGA NPs and/or EGCG-KE NPs) was removed using a spin coater. How did you control the dosage amount?  

4. Mention the loading capacity of your nanoparticles or at least the drug entrapment percentage?

5. Mention the scale bar of figure 3 A and B.

Author Response

Please see the attachment (Reviewer 3).

Reviewer #3

Comment: 1. Correct the in text references throughout the paper.

Response: We thank the reviewer for the close read and suggestions. We have checked the entire manuscript carefully and corrected format of references to follow the guidelines of the journal.

Comment: 2. Higher doses of EGCG are hepatotoxic, if 750 mg/kg dosed twice a day (doi: 10.1016/j.fct.2009.10.030). Daily intake of 800 mg or more could increase risk of liver damage (doi:10.2903/j.efsa.2018.5239). Did you consider the toxicity concentration levels of EGCG and the dosing amount when designing this study?

Response: Thank you for raising this point. We have considered the dosing of EGCG from relative references. In our study, our dosage of EGCG loaded in microneedles was 4 μg a week (nearly 160 μg/kg in transdermal way). It’s more less than toxicity dosing (750 mg/kg in oral way).

Compared with oral administration and injection, microneedle transdermal systems are able to reduce systemic toxicity and promote precise drug efficiency in lower dosage. Our study showed that 4 μg of EGCG a week by microneedle transdermal system could promote hair shaft density.

Comment: 3. In section 2.3, it is mentioned that the PDMS cathode mold is prepared by the template method. PVP solution (1 g/ml) mixed with the drug (RAPA-PLGA NPs and/or EGCG-KE NPs) was added to the PDMS microneedle mold and then vacuumed for 2 min. The excess PVP solution (assuming it contains RAPA-PLGA NPs and/or EGCG-KE NPs) was removed using a spin coater. How did you control the dosage amount? 

Response: We thank the reviewer for pointing this out. The method of controlling the drug dosage amount was shown as followed (Figure 5).

First, PVP solutions contains different amounts of RAPA-PLGA NPs and/or EGCG-KE NPs were prepared. For example, 1 g of PVP was mixed with 1 ml of RAPA-PLGA NPs solution(containing 1 mg RAPA in NPs). Then the microneedle wells (2 μl/MN patch) were filled with PVP solution (0.5 mg/ml RAPA) by vacuumed (step 1). The excess PVP solution were removed by spin coater to ensure only wells containing drugs(step 2).Thus the microneedle wells contains 1 μg(2 μl ×0.5 mg/ml) of RAPA. The PVA solution(green) were coved as a substrate (step 3). In this way, a microneedle patch (1 μg of RAPA) was prepared. Different dosages of drug amount were controlled as the same way. In addition, actual dosages of microneedle patches were measured. It demonstrated that the testing actual dosage is 1.084±0.053 μg, showing small error.

Figure 5.Steps to prepare microneedle patch (see attachment)

Comment: 4. Mention the loading capacity of the nanoparticles or at least the drug entrapment percentage?

Response: Thank you for your comments. The loading capacity of the nanoparticles were mentioned in line 193 and 194 as followed : " The loading capacity of RAPA and EGCG NPs was 82% and 76%, respectively. "

Comment: 4. Mention the scale bar of figure 3 A and B.

Response: Thank you for raising this point. The scale bar of figure 3 A and B were mentioned in line 198 and 199 as followed: " SEM image of PLGA-RAPA NPs (A) and EGCG-KE NPs (B). Scale bar: 200 nm. "
